# Household Groups’ Land Use Decisions Investigation Based on Perspective of Livelihood Heterogeneity in Sichuan Province, China

**DOI:** 10.3390/ijerph19159485

**Published:** 2022-08-02

**Authors:** Hong Tang, Jian Liu, Xiaowen Dai, Yun Zhang, Wendai He, Qi Yin, Feng Huang, Ruiping Ran, Yunqiang Liu

**Affiliations:** 1School of Management, Sichuan Agricultural University, Chengdu 611130, China; tanghong@sicau.edu.cn (H.T.); liujian@stu.sicau.edu.cn (J.L.); daixiaowen@sicau.edu.cn (X.D.); zhangyun@stu.sicau.edu.cn (Y.Z.); hewendai@163.com (W.H.); yinqi@sicau.edu.cn (Q.Y.); ranruiping@sicau.edu.cn (R.R.); 2Sichuan Rural Development Research Center, Chengdu 611130, China; 3School of Economics, Sichuan Agricultural University, Chengdu 611130, China; 72038@sicau.edu.cn

**Keywords:** livelihood capital, livelihood diversity, livelihood strategy, livelihood heterogeneity, land use decision-making, household group, Sichuan

## Abstract

Land use decision-making is a vital livelihood strategy associated with the rational collocation of livelihood asset endowments by rural households. Based on the perspective of livelihood heterogeneity, this paper collected the data from 540 farm households in 27 villages in three Sichuan Province counties to identify the land use decision-making characteristics of the household groups. A land use decision-making framework (LUDF) based on the sustainable livelihood framework (SLF) was established and dynamic and dual indicators were developed to divide the sample into six household groups. The household livelihood capital, livelihood strategies, and livelihood diversity were then analyzed at the regional and household group level, and the land use decisions of these household groups were explored, from which the following was found. (1) Overall livelihood capital in the study area was low, and except for human assets, there were few other assets, with households in the survey areas being more inclined to engage in non-farming livelihood activities; however, households in Nanjiang and Qionglai had greater livelihood activities choices than Luxian. (2) The LL-type household was the chief household group; the household group distribution in Qionglai was well-proportioned but uneven in Nanjiang and Luxian; and the HL-type, ML-type, and LL-type household livelihood strategy choices were polarized. (3) Most households were involved in land self-cultivation, followed by land transfer-in, land transfer-out, and land abandonment households. Specifically, there were more households that cultivated fragmented landholdings than specialized households with large-scale landholdings, the land transfer rate was relatively low, the transfer-in land area was far greater than the transfer-out land areas, and a small number of households that had abandoned their land were still involved in agricultural production. (4) There were obvious discordant human–land relationships and inefficient land uses in the study area. Based on these findings, relevant policy recommendations are given to improve farm household livelihood capital, optimize livelihood strategies, and assist in land use decision-making.

## 1. Introduction

Individual rural households are widely considered to be the basic economic management organizational units and indispensable decision-making bodies for land use in rural areas around the world [1]. Whether used in production or life, agricultural land is closely correlated with rural households [2]. However, because of urbanization, industrialization, and agricultural modernization, rural households’ decision-making about land use has gone through extensive far-reaching changes. In China, the rural population decreased by 244.36 million from 2000 to 2018, with much of this population moving to urban areas [3]. This directly resulted in a weakening of the links between the rural household and agricultural land, thus leading to rural hollow, rural decline, non-farm use of land, and land abandonment [4]. Concurrently, considerable stay-at-home elders and women are still cultivating the small fragmented contracted land to meet the survival or development demands, which has resulted in a significant increase in the rural aging population [5] and serious land fragmentation [6]. Nevertheless, the growing imbalance between the rural population and agricultural land has also aroused frequent land transfers, which have promoted the moderate scale operation of agricultural land, encouraged mechanized production, increased labor force productiveness, and reduced agricultural operating expenditure [7]. Therefore, paying attention to rural households’ land use decisions have naturally become of interest to both governments and academia. Numerous previous studies on household land use decisions have tended to focus on single land use decisions, such as land transfers [8] or land self-cultivation [9], but there have been relatively few studies exploring land abandonment [10] or multiple land use decisions, such as land transfer-in, land transfer-out, land self-cultivation, and land abandonment.

Livelihood capital, which is the basic means of survival and development for rural households and is vital in helping to resist natural disasters, reduce poverty, and enhance livelihood sustainability and adaptability [11], has a significant influence on rural household livelihood strategies and livelihood outcomes as well [12]. Agricultural land, as the important assets utilized by farm households, provides them with an income source, food, and social security [13]; however, the planting structures, the fund and labor force investment, the land utilization efficiency, new techniques, and final production output are inevitably determined by livelihood capital [9]. The highly diverse employment channels resulting from economic development and technology advancement have promoted rural households to have a hand in multiple employment choices and stimulated them to adjust their land use decisions for the optimal benefit. Thus, it is of great significance to explore and understand the characteristics and patterns of rural household land use decision-making based on the perspective of livelihood capital and livelihood diversity, which could be conducive to understanding the causal pathway of households’ land use decision-making.

The sustainable livelihoods framework (SLF) provides a comprehensive, logical analysis perspective with which to examine the interrelations between livelihood capital, livelihood strategies, and livelihood outcomes [14]. In recent years, previous studies have applied the SLF to identify the affect factors impacting livelihood strategies [15]; assess livelihood sustainability [16], vulnerability [17], adaptability [18], and stability [19]; and explore the constraints on and incentives for rural household livelihoods [20], from which quite meaningful research accomplishments have been achieved. The SLF has also been used to examine and analyze the interactions between the household livelihood capital and the land use decisions of selected subjects living in ecologically fragile rural areas, such as the Himalayas in Nepal [21], Indonesia [22], Ethiopia [23], Ningxia [24], and Yunnan [25], which are all located in mountainous areas in China. Some studies have also focused on suburban areas around metropolises [26], ethnic symbiosis areas [27], agro-pastoral areas [24], and migration resettlement areas [28]. Specifically, most previous studies have focused on the planting structure, the plots and areas of land, fertilizer management, land transfers, and the land inputs and outputs, etc. By observing these findings through the SLF under a changeable natural and social background, we can have a deeper understanding of the land use logic behind heterogeneous livelihood capitals. Thus, by identifying and improving the key analysis pathways of the SLF, it is possible to identify the interconnections between livelihood capital, livelihood strategies, and the land decision-making process.

In addition, it is necessary to conduct the study of livelihood capitals and land use decision-making at differentiated dimensions and scales in order to accumulate a highly rich experience in coping with the constant turbulences and challenges from society, the economy, and the climate. However, existing studies at the individual [29], community, and region level [30,31] have rarely focused on household groups or have only divided these groups using a single indicator, such as livelihood capital or income sources [32]. Therefore, examining the livelihood capital, livelihood strategies, and livelihood diversification of rural household groups using diverse measures; determining the patterns and characteristics of correspondent household groups; and identifying the specific correlations require deeper research.

China is a large developing country that has experienced significant social reforms and ecological environment changes since the implementation of the Reform and Opening-up Policy in the late 1970s. Millions of rural households have been influenced by national policies and strategies and are more or less passive in adjusting their livelihood strategies in response to the radical socio-economic development [33]. In 2017, a rural revitalization strategy was proposed in the 19th Party Congress report with the purpose of solving the quandary about rural production and life as early as possible [34]. Sichuan Province, known as a large representative agricultural province in China, had a substantial but unevenly developed rural population of around 36.21 million in 2020 [35]. The urbanization rate in Sichuan was close to 60% in 2020, which was similar to the national level of 63.9%. The total grain output of Sichuan in 2020 was 35.27 million tonnes or 5.3% of the national output [36]. Sichuan Province is a large grain-producing, agricultural province and plays a key food security role in China; however, the province has extremely inefficient land use, a high degree of land abandonment, small-scale land operations, and incomplete land transfers [37]. However, as the rural households in Sichuan need to adjust their livelihood strategies related to land use, labor force, and employment transformation decisions, due to their typicality and representativeness, research on rural households and their land use decisions in Sichuan Province could be valuable in informing future rural policies.

This study randomly sampled Qionglai, Nanjiang, and Luxian counties in Sichuan Province, from which valid data were gathered from 540 households. The main objectives of this study were as follows: first, to develop a logical theoretical analysis framework to examine the causal pathways and correlations between livelihood capital, livelihood strategy, household group, and land use decision-making; second, to quantify the value of the livelihood capital and livelihood diversity indexes, draw upon these dual indicators to classify household groups, explore the livelihood capital, livelihood strategy, and livelihood diversification characteristics, and analyze the household land use decision-making patterns and characteristics based on livelihood heterogeneity; third, to develop a reference for the implementation of policies to promote rational and efficient land use, the orderly circulation of land, enhance agricultural production, improve household living conditions, and optimize human–land relations.

## 2. Materials and Methods

### 2.1. Theoretical Framework

Differentiated household types, which are household groups rather than individual households, are determined and easily influenced by their intrinsic livelihood endowments. They tend to adopt relatively coherent land use decisions to achieve profitable and steady livelihood outcomes. Investigating household land use decision-making processes from a livelihood capital perspective can provide a deeper understanding of the observed phenomenon and the underlying mechanisms. The SLF, proposed by DFID [38], has been widely used as the analysis pathway for examining household livelihood capital and land use behaviors [39,40]. In particular, five rural household livelihood capital dimensions, human, natural, physical, financial, and social, are the main SLF concerns for the preservation of sustainable livelihoods, reductions in livelihood vulnerability, and the enhancement of livelihood adaptability. Livelihood capital not only directly affects household livelihood strategy choices but can also be indirectly employed as a measure for classifying different household groups and exploring household group land use decision patterns and characteristics.

Apart from the influence of livelihood capital on hindering or driving livelihood changes [19], it is also affected by external and internal factors [41]. Livelihood capital and livelihood strategies can be significantly affected by external factors, such as national policies, intractable socioeconomic issues, and unexpected natural disasters [42]. For example, the Household Contracted Responsibility System (HCRS), which was implemented in 1978, contributed to improvements in the possession and allocation of disposable livelihood capital, especially for essential household natural assets such as contracted land, but also prompted the surplus agricultural labor force to enter non-agricultural industries [43], which reversed the straitened circumstances of the vulnerable and single livelihoods of rural households [44]. After entering the 21st century, rural households were significantly affected by local and global urbanization and industrialization [45,46]. The increase in non-farming income resulted in a lot of young people leaving the countryside and abandoning farming (such as abandoning or transferring-out land). Therefore, the livelihood capital, livelihood strategies, and relationships with the land of this new generation changed significantly. Consequently, the livelihood capital and livelihood strategies of the household types became more differentiated as households adopted different land use decisions to adapt to their special situations.

Because of the influence of internal micro-economic mechanisms (rationality) or other factors (cultivation traditions and local culture), rural households could be regarded as rational decision units that configure their livelihood capital to conform with their livelihood strategies and land use policies, with these different land use decisions, such as land transfer-out, transfer-in, self-cultivation, and abandonment, exerting homologous influences on livelihood capital and other livelihood strategies, which result in heterogeneous livelihood outcomes and reveal the interactive processes. Therefore, “land use decisions” are introduced as an independent part of this theoretical framework. The original SLF logic was improved and adopted to align with the developed LUDF (land use decision-making framework), which reflects the above underlying effect mechanisms, as shown in Figure 1.

### 2.2. Study Area and Data Sources

Sichuan Province is located in the hinterland of southwest China, where the terrain is high in the east and low in the west. Sichuan has a complex, diverse topography with plains, mountains, and hills. In China, Sichuan Province is a highly populated province and is an important economic, political, and cultural region. Therefore, the rural population in Sichuan Province is highly typical (Figure 2). The specific survey process and data sources were as follows.

To select the representative samples, three prefecture-level cities were first selected: Chengdu, Luzhou, and Bazhon. Then, using a random stratified sampling method based on the economic development level differences, three sample counties, one from each prefecture-level city, were randomly selected, Qionglai, Luxian, and Nanjiang, based on their economic development situations, locations, and populations. After this, three sample townships were identified in each county, that is, a total of nine sample townships. Second, based on the economic development level in the villages, the geographical transportation, the population density, and the quantity and quality of the cultivated land, 3 villages were randomly selected from each sample township; therefore, 27 sample villages were identified. Third, based on the rural household random number tables obtained from the village committees, around 20–25 rural households were randomly selected as the subjects from each sample village.

Forty-two students with previous experience in similar studies were selected as formal surveyors, who were then divided into three teams, each of which was assigned an extra instructor to assist with the survey process. In addition to the instructor, 11 of the 14 investigators in each group were responsible for the household surveys, and the other three investigators were responsible for checking and reviewing the quality of the questionnaires and as substitute investigators when needed. Before the formal investigation began in September 2021, all investigators were trained for 10 days to ensure they had a good understanding of the questionnaire. To improve the draft questionnaire, it was pilot tested on 30 households, after which 15 investigators were randomly selected from the three teams to discuss issues with the questionnaire with the village heads, the village party secretaries, and the village accountants before the formal survey. The group discussion was conducted to improve the design logic of the questionnaire.

To ensure survey authenticity, before the interview, the households were informed about the “Privacy Statement”, which stated that the views and answers provided would only be used for scientific study and that all information would be kept private. To reduce possible information and data bias, content fairness and justice were focused on in the questionnaire design. The surveyors were trained to pay attention to the question wording and to avoid any wording that could inhibit the giving of authentic answers. To encourage the respondents to be involved, 4 kg of edible oil was given to each respondent as a gift before the interview.

In October 2021, based on the PRA (Participatory Rural Assessment), the three teams from the Sichuan Center for Agricultural Development Research conducted the questionnaire and semi-structured interviews with the selected rural households. The average interview time for each rural household was between 2.5 and 3.5 h. The questionnaire gathered information on the basic situation in 2020 in each rural household, such as their production, assets, income, expenditure, and land use decisions. As a consequence of gaining 180 valid questionnaires per county in order to ensure the comparability among regions, a total of 540 valid questionnaires were obtained.

### 2.3. Measurements and Methods

#### 2.3.1. Livelihood Capital Index System

Livelihood capital refers to the survival support resources owned by rural households to survive livelihood vulnerabilities and risks. Referring to the livelihood capital classification setting of the Sustainable Livelihood Analysis Framework (SLA) [38], combining with the actual situation in the survey area and related existing studies of Ding et al. [47] and Wang et al. [48], this study identified five rural household livelihood capital dimensions: human capital, natural capital, financial capital, physical capital, and social capital. Human capital, as an essential contributive factor to rural household survival and development, refers to the household’s human resources, which are measured by the proportion of the labor force to the total population, per capita education attainment, and per capita degree of health. Natural capital refers to the natural resources owned by households, the per capita cultivated land area, the per capita orchard land area, and the per capita forest land area. Financial capital refers to the capital and related funds that rural households acquire and accumulate from production and life, per capita annual income, per capita cash and bank savings, and per capita loan funds. Physical capital refers to the fixed capital with a certain value that is utilized by the households to assist their livelihood, per capita fixed assets, per capita standardized housing area, and per capita livestock. Social capital refers to the social network between the household and others, that is, the people available to offer aid when necessary, which was measured by the household members in public office and the per capita expenditure from maintaining social relations. The specific livelihood household capital evaluation index system is shown in Table 1.

The livelihood capital weights were assessed based on the studies of Zhang et al. [49] and Yuan et al. [50]. While the analytic hierarchy process (AHP) method has been widely used to judge the importance of subjects, it can be subjective, and the corresponding results can be disturbed by artificial factors. However, the entropy method is objective, and the biases can be corrected to a certain extent. Therefore, a combination of the entropy method and AHP method was used to determine the comprehensive weights for each of the five livelihood capital dimensions, after which a minimum–maximum standardization method was used to normalize the data to obtain the standardized value for each livelihood capital index. The final household livelihood capital score was determined by multiplying the livelihood capital index weight with the corresponding normalized livelihood capital value, with the total household livelihood capital score being equal to the sum of the five dimensional livelihood capital scores, the formula for which was as follows:(1)$$L=\sum_{i=1}^{5}\sum_{j}^{m}W_{ij}X_{ij}$$
where L was the total score for the livelihood capital of a certain household, i was the i-th dimensional livelihood capital, j was the j-th livelihood sub-indicator for the i-th dimensional livelihood capital, W_ij_ was the combined weight of the j-th livelihood sub-indicator, and X_ij_ was the household’s normalized livelihood value for the j-th livelihood sub-indicator.

#### 2.3.2. Livelihood Strategy Classification

As in the previous research of He et al. [51] and Li et al. [52], and based on the actual situation in the survey area, this study used the proportion of non-agricultural income to total income as the primary basis for dividing the rural household livelihood strategies. The specific division method for household types was as follows: first, the household was distinguished by whether it was only engaged in agricultural activities, after which the livelihood strategies were divided into pure-agriculture, part-time, and non-agriculture. Then, the ratio of part-time household non-agricultural income to total income was calculated (over 90%, between 50 and 90%, between 10 and 50%, and lower than 10%), which in turn was respectively classified as non-agriculture, non-agriculture-dependent, agriculture-dependent, and pure-agriculture livelihood strategies.

#### 2.3.3. Livelihood Diversity Index Calculation

The livelihood diversity index reflects the diversity of household livelihood activities. However, because of the significant differences in the multiple livelihood activities in rural households, taking the livelihood strategy as a measure does not fully represent the richness of specific rural household livelihood activities. Therefore, based on the highly complicated income sources identified from the rural household questionnaires in the survey area (Table 2), the Simpson index method was employed to identify the household livelihood diversity indexes, measure livelihood activity diversification, and divide the rural household types. The formula was as follows:(2)$$K_{j}=1-\sum_{i}^{m}T_{\mathrm{ij}}^{2}$$
where K_j_ was the livelihood diversity index of the j-th household, i was the j-th household’s i-th subdivision livelihood income, T_ij_ was the ratio of the j-th household’s i-th sub-division income to total income, and m was all subdivision income sources. The value of K ranged from 0 to 1: the greater the value of K, the more diverse the household’s livelihood strategies, and vice versa. When K = 0, the rural household was engaged in a single or specialized livelihood activity. For example, when rural households had transferred-in a large amount of arable land (maybe more than 10 ha) and invested mostly in a family farm, it was probably regarded as a specialized production or a specialized livelihood activity.

#### 2.3.4. Household Group Classification

Previous studies mostly determined rural household types from the ratio of non-agricultural income to total income [53]. This study, however, did not identify the livelihood strategies using the non-agricultural income ratio because classifying household types using a single indicator does not reflect the diversity of household livelihood strategies. In reference to the previous household type division in Wang et al. [39] and Zhang et al. [49], and accounting for the actual situation in the survey area, the double combined indicators livelihood capital value and livelihood diversity index were adopted to classify the household types. The detailed division steps were as follows. First, by comparing the individual household total livelihood capital value with the average total household livelihood capital value in the survey area, the households were divided into high-level and low-level livelihood capital household types. Then, the individual total livelihood capital values in the high-level livelihood capital households were compared with the average total high-level livelihood capital, from which the medium and final high-level livelihood capital household types were determined. Finally, the low-level, medium-level, and high-level livelihood capital household types were determined. Similarly, the households were classified into high-level and low-level livelihood diversity household types based on the livelihood diversity index. Finally, by cross-combining the above household types based on livelihood capital and livelihood diversity, a total of six farm household types were determined: HH-type (high livelihood capital and high diversity households); HL-type (high livelihood capital and low livelihood diversity households); MH-type (medium livelihood capital and high livelihood diversity households); ML-type (medium livelihood capital and low livelihood diversity households); LH-type (low livelihood capital and high livelihood diversity households); and LL-type (low livelihood capital and low livelihood diversity households).

## 3. Results

### 3.1. Livelihood Capitals

The household livelihood capital values in the different study areas are shown in Table 3. On the whole, the human capital value was the highest, and the natural capital, financial capital, physical capital, and social capital were relatively low. The subsequent analysis results for the livelihood capital outcomes were as follows.

#### 3.1.1. Human Capital

The human capital’s objective richness reflects the livelihood capabilities of rural households. Table 3 shows that the human capital of the households in the study area was high, with an average value of 0.152, which was far greater than the other capitals. The household human capital in Luxian was the lowest and in Qionglai was the highest, which suggested that there were obvious human capital differences between the households in the different regions, perhaps because economic and education development differences among the regions led to this situation.

#### 3.1.2. Natural Capital

Rural households can achieve corresponding profits by investing in their labor force and funding and managing their natural capital. It can be seen from Table 3 that the overall natural capital of the households in the study area was the lowest, with an average value of only 0.02, which was much lower than the human capital. This might indicate that the natural resources in the study area were scarce. Therefore, the dependence on natural capital by rural households was possibly much weaker than that on the other capitals, especially human capital. In addition, probably due to households’ non-agricultural livelihood choices, they paid less attention to managing natural assets, which resulted in the depreciation of natural resources.

#### 3.1.3. Financial Capital

Financial capital can strengthen livelihood capability and improve poor living conditions. There was relatively scarce financial capital in the study area, at an average value of 0.007, which was only slightly higher than the natural capital and less than one-tenth of the human capital. This may have been because the fluctuations in outside household working income were affecting their financial capital accumulation.

#### 3.1.4. Physical Capital

When households carry out specific livelihood activities in agriculture and their normal lives, physical capital is an indispensable physical foundation. Although the household physical capital in the study area ranked second, it was much less than the human capital. The household physical capital in Qionglai was slightly higher than the average (0.014), but in Luxian it was the lowest. Relatively insufficient rural household physical capital could hinder everyday livelihood activities and the upgrading of livelihood strategies.

#### 3.1.5. Social Capital

The social capital in Luxian was lower than the mean value of 0.011 in the study area, which indicated that there were fewer available social network resources. Even though the household social capital in Qionglai and Nanjiang was relatively good, the overall level was still weak, perhaps because of households’ stronger dependence on the economic contract for survival and development, rather than human relationships in the study area.

### 3.2. Livelihood Strategy and Livelihood Diversity

#### 3.2.1. Livelihood Strategy

The household livelihood strategies were divided into four types based on the ratio of household non-agricultural income to total income. As shown in Figure 3, non-farming and non-agriculture-dependent livelihood activities were common. The percentage of rural households engaging in non-agricultural livelihood strategies was as high as 56.7%, and the percentage that was non-agriculture-dependent was 19.4%, which suggested that there was a significant number of households participating in non-farming employment. Contrarily, there were significantly fewer households with agriculture-dependent livelihood strategies at around 6.3%.

Figure 4 shows that the ratio of livelihood strategies in the different study areas was generally non-agriculture > non-agriculture-dependent > pure-agriculture > agriculture-dependent. The number of households in Qionglai with pure-agriculture livelihood strategies was second only to non-agricultural livelihood strategies and non-agriculture-dependent livelihood strategies, which was inconsistent with other regions and indicated that Qionglai households may be more positive about agricultural production and operations than those in Luxian and Nanjiang.

On the whole, due to the continual progress in the optimization of both external and internal conditions, rural households have been more inclined to adjust their labor force and assets allocations in line with their development interests, that is, realizing all-round development to receive optimal benefits. This was generally consistent with the general Chinese population trends of urbanization and the non-agriculturalization of production. Therefore, in comparison to agriculture production, non-agriculture employment was possibly easier and more economically attractive for most rural households to be involved in.

#### 3.2.2. Livelihood Diversity Index

The household livelihood diversity index was calculated based on the household income ratios of the various livelihood activities to total income (Table 4). As shown in Table 4, the household livelihood diversity indices in Qionglai and Nanjiang were both relatively high and close, while in Luxian, it was lower than the mean value (0.215). These results indicated that the potential household livelihood opportunities in the different study areas were different; that is, there were discrepancies in household livelihood diversity. It may be that livelihood activity opportunities were partly affected by economic development. Therefore, households living in economically poor areas were more likely to be limited in taking employment, thus leading to less diverse livelihood activities.

### 3.3. Household Types

#### 3.3.1. Household Type Analysis

A combined dynamic method based on livelihood capital and the livelihood diversity index was employed to subdivide the investigated households into six types, after which the livelihood capital, livelihood diversity, and livelihood strategy differences and commonalities were examined. The specific analysis results were as follows.

All six rural household ratio types are shown in Figure 5. The highest percentage was found to be the LL-type rural households at 34.3%. There were somewhat fewer LH-type and ML-type rural households at 17.4% and 16.1%, and even fewer HH-type and ML-type rural households at 10.7% and 12.2%. The percentage of HL-type rural households was the lowest at 9.3%. These results revealed that there was an overall livelihood capital deficiency and low-level livelihood diversity in the survey area.

By analyzing the household type distributions in the different study areas (Figure 6), it was found that the household type orders in Nanjiang and Luxian were: households with low livelihood capital (102 and 130) > households with medium livelihood capital (46 and 35) > households with high livelihood capital (32 and 15). The different household type distributions had large deviations, with most LL-type households having low-level livelihood capital and livelihood diversity indices. However, in Qionglai, the household type order was: households with medium livelihood capital (72) > households with high livelihood capital (61) > households with low livelihood capital (47). Therefore, the household type distributions were relatively well-proportioned in comparison to Luxian and Nanjiang.

Because Qionglai has a flat topography, convenient traffic, and superior irrigation conditions, and as it is extremely close to the Chengdu metropolitan area, there is rich potential for employment in the non-agricultural sectors. As households can manage their agriculture or enter secondary or tertiary industries, they generally had high-level livelihood capital and diverse livelihood choices, as seen in the higher distributions of HH-type and HL-type households in Qionglai. In contrast, Nanjiang and Luxian are mountainous and hilly, relatively far from the metropolis, and have limited regional development resources. Therefore, as there were far fewer non-agricultural employment opportunities in Luxian and Nanjiang, the household livelihood capital and livelihood diversity were weak; therefore, there was a greater number of LH-type and LL-type households.

#### 3.3.2. Group Level Household Type Analysis

The household group results for the primary livelihood characteristics, livelihood capital, livelihood diversity index, and livelihood strategies for the six household types are respectively shown in Table 5 and Table 6 and Figure 7.

(1)HH-type households. There were 58 HH-type households with a total livelihood capital value of 0.2735, which was second only to the HL-type households (0.2815). The human capital in the HH-type households was the highest, followed by physical capital and social capital; however, the financial capital and natural capital were relatively weak. The per capita education attainment (9.47 years) and livelihood diversity index (0.484) were the richest, and the annual income per household (118,300 CNY) was fairly high, but the per capita cultivated land area (0.099 ha) was much less than for the HL-type households (1.477 ha). As the HH-type households had rich human capital, they tended to leave agriculture to actively explore diverse non-agricultural employment channels and pursue higher non-agricultural income sources, which was inconsistent with the single and large-scale agricultural production activities.(2)HL-type households. There were 50 HL-type households, which had the highest total livelihood capital of all six types (0.2815). HL-type households were characterized by rich human capital; medium-level financial capital, physical capital, and social capital; and weak natural capital. Notably, the HL-type household annual per household income (174,400 CNY), the per capita cultivated area (1.4774 ha), and the per capita forest area (0.1554 ha) were higher than in the other household types. However, the HL-type livelihood diversity index (0.0405) was much lower than the HH-type, MH-type, and LH-type households. This suggested that HL-type households relied on their human livelihood capital, physical capital, and social capital advantages and chose to engage in specialized and large-scale agricultural production activities. However, some HL-type households also transferred into the non-agricultural sector to participate in secondary and tertiary industries. Generally speaking, the HL-type households had a moderate tendency to leave the agricultural sector, and there appeared to be a polarizing trend in their choice of livelihood strategies.(3)MH-type households. There were 87 MH-type households, with a total livelihood capital value of 0.2108. MH-type households had high human capital and middle-level physical capital and social capital; however, they had low natural capital and financial capital. The MH-type households livelihood strategy ratio was non-agriculture > non-agriculture-dependent > pure-agriculture > agriculture-dependent. The labor force per household was 3.08, which was second only to the HH-type households. Notably, the MH-type households owned the largest per capita orchard area (0.1621 ha) but the least per capita cultivated land (0.0791 ha), which suggested that the human–land separation in MH-type households was high, as a higher labor force was needed for the cultivated land than for the orchard or forest land. The MH-type household livelihood strategies were more flexible and diversified, with a livelihood diversity index of 0.454, and they also had a low-level livelihood risk.(4)ML-type households. There were 66 ML-type households, with a total livelihood capital value of 0.2092, which was slightly lower than in the MH-type households. The ML-type households had rich human capital, followed by physical capital; however, the other capitals were relatively weak. Only a small percentage of ML-type households had non-agriculture-dependent and agriculture-dependent livelihood strategies, which suggested that the household livelihood strategy choices were characterized by a polarization toward the pure-agriculture or the non-agriculture sectors. The per capita cultivated area (0.3102 ha) was second only to the HL-type households, and the per capita standard housing area (67.91 m^2^) was the largest of all household types. However, the ML-type household livelihood diversity index was exceedingly low (0.0336), indicating that they had limited livelihood strategy choices.(5)LH-type households. There were 94 LH-type households with a total livelihood capital value of 0.1509, which was the second lowest of all household types. While they had relatively rich human capital, the other capitals were quite low. The LH-type household labor force per household (2.15), per capita education attainment (5.75 years), and annual income per household (67,800 CNY) were also second lowest; their per capita areas of cultivated land (0.084 ha), forest land (0.027 ha), and orchard land (0.008 ha) were relatively small; and they were more inclined to move away from agricultural production. Although the LH-type households had many and varied livelihood strategies (0.4205) because they had diversified into non-farm livelihood activities, most of their jobs in the secondary and tertiary industries were stable but had low-level average incomes.(6)LL-type households. There were 185 LL-type households, which was the highest of all household types. This group had the lowest total livelihood capital value of 0.1319, with all other capitals also being relatively low. In comparison to the other household types, the LL-type household labor force per household (1.16), annual income per household (47,200 CNY), and per capita education attainment (5.20 years) were weak. Because of the weak and insufficient livelihood capital and the low human capital, their livelihood strategy choices were limited, which meant that increasing their incomes was difficult and gave them fewer opportunities to improve their livelihood resources and enhance their poor living conditions. The polarization trend in the LL-type household livelihood strategies was relatively obvious.

### 3.4. Decision-Making on Household Land Use

The households in the different regions made heterogeneous land use decisions. Table 7 shows the land use decisions in the different households, and Table 8 shows the ratio of the different household land use decisions. It was found that the land self-cultivation, land transfer-out, land transfer-in, and land abandonment proportions in the different households were respectively high, medium, medium, and low. Because of the differences in the quality of the cultivated land, the farming facilities, and the locations, the land transfer income and expenditure differences in the regions ranked Qionglai > Nanjiang > Luxian. On the basis, households in Nanjiang and Luxian were more inclined to rent in land, and the land transfer-in scale was a little higher than the land transfer-out scale. Households in Qionglai were more inclined to transfer-out land, perhaps because of higher rent (10,224 CNY/ha per year); however, due to the superior production conditions for the moderate scale of operation, the transfer-in land scale was much greater than the transfer-out land scale. Quantitatively, more households (38) in Nanjiang were abandoning land than in the other regions and also had the largest abandonment land area (0.25 ha) and the largest self-cultivation land (0.28 ha), perhaps because of the lower land quality and production level. Luxian had the largest number of self-cultivation households (86), showing that the smallholder operation of land was still prevailing in Luxian.

#### 3.4.1. Land Transfer-Out

Table 9 shows the household land transfer-out characteristics. The 143 households that transferred land out, which accounted for 26.5% of all households investigated, had an average land area of 0.20 ha per household. Table 8 indicates that the richness of the household livelihood capital may have affected the land transfer-out decisions. Because of the rising livelihood household capital, the household type transfer-out land proportions were HH-type > HL-type > ML-type > MH-type > LH-type > LL-type. ML- and MH-type households with the same livelihood capital levels but different livelihood diversities had few transfer-out land differences; however, the transfer-out land areas between the high-level and low-level livelihood households had some differences, which indicated that the livelihood diversity differences weakly influenced household land transfer-out behaviors.

Because the HH-type and HL-type households had a high social capital and developed social networks, they were better able to rent their land to relatives, large planters, or people in the same village, which provided relatively high annual land rents (10,776, 10,328 CNY/ha per year, respectively). The MH-type and ML-type households mainly transferred their land out to big planters or other people in the same village and received a medium-level annual rent (9463, 9761 CNY/ha per year, respectively). The LH-type and LL-type households tended to rent their land to big planters, other people from the same village, people from other villages, and agricultural enterprises as potential tenants and received the least rent (7418, 8567 CNY/ha per year, respectively). These results implied that the differences in livelihood capital had some effect on land tenant selections and the rents that could be charged.

The reasons for the household land transfer-outs varied. While the HH-type and HL-type households had richer human capital, the lack of insufficient labor appeared to be the main reason for the land transfer out, which also indicated that the labor force in these households was possibly more inclined to enter non-agricultural activities. Interestingly, no HH-type and HL-type households claimed that poor land quality was the main reason for the land transfer-out, which indicated that because of the availability of non-agricultural activities, the agricultural labor force had been dropping sharply in rural families, which had indirectly resulted in a decline in the households’ enthusiasm for farming superior land. This, in turn, intensified the human–land separation and encouraged them to rent out land. The land transfer-out driver in the MH-type and ML-type households appeared to be a lack of farming labor force, a uniform land transfer-out trend inside the village, and low income from the land. Insufficient labor, uniform land transfer-out inside the village, and low land income were also the main reasons for land transfer-out in the LH-type and the LL-type households. It possibly suggested that the livelihood capital differences had a weak effect on middle-level and low-level livelihood households’ land rent-out decisions.

#### 3.4.2. Land Transfer-In

Table 10 shows the household land transfer-in characteristics. The 159 households that had made land transfer-in decisions accounted for 29.4% of the investigated households, with the average rent-in land area per household being 2.48 ha, which was far greater than the transfer-out land per household (0.20 ha) and indicated that the households with transfer-in land preferred to manage comparatively large-scale and specialized agricultural land. Compared with the average annual land transfer-out rent (6537 CNY/ha), the average annual land transfer-in expenditure (7896 CNY/ha) was lower. Table 8 shows the transferred-in land household proportions to be HL-type > LH-type > LL-type > ML-type > MH-type > HH-type. In the high- and medium-level livelihood capital households, there were significant transfer-in land area differences; however, the differences between the low-level livelihood households were not obvious. It may be because households with richer livelihood capitals were likely to pursue more surplus values of agriculture operation. Therefore, in comparison to the other household types, the HL-type households, as the biggest agricultural land operators, had exceedingly rich human capital; relatively high social capital, physical capital, and financial capital; and were renting the most land at 13.65 ha per household. Nevertheless, as their specialized and single livelihood activities could be easily affected by natural disasters, agricultural product price fluctuations, and other related unpredictable factors, their livelihood risks were possibly greater.

The HH-type households grew more non-food crops than food crops, and the HL-type households equally grew food crops and non-food crops; however, the other household types’ transfer-in land was mainly used for food crops, followed by non-food crops. This illustrated that the main reason driving high-level livelihood capital households to favor non-food crops was possibly because the prices were better for general food crops and there was a sizable added value attached to cash crops. Therefore, these households were willing to invest in land, advanced technology, and their labor force, which required a large initial injection of funds, and the willingness to undertake certain business risks to expand the agricultural production profit margins. However, the medium-level and low-level livelihood capital households mostly chose to plant food crops. Two potential inferences were made from these results. First, there was a reduction in the labor force costs and initial funds for agricultural operations, and second, households sought to obtain relevant agricultural policy subsidies to avoid market risks and receive a certain level of operating income.

There were several reasons for households to transfer-in land. The main reasons appeared to be requests by relatives and friends to manage land and the sizable economic benefits to be gained from the land. Because of their richer human capital, the proportion of HH-type, HL-type, and MH-type households with sufficient labor that were motivated to transfer-in land was slightly higher than in the other household types. Other reasons for transferring-in small quantities of land were dependence on farming skills for living, the need to provide feed for livestock, or the need to satisfy daily food needs. Therefore, the reasons for the land transfers-in were diverse. Due to the differences in livelihood capital, the high livelihood capital households who transferred-in large scale land were perhaps pursuing economic interests; however, the reason low livelihood capital households transferred-in small amounts of land was possibly to meet the requests of their friends and relatives.

#### 3.4.3. Land Self-Cultivation

Land self-cultivation refers to managing and operating contracted land without land transfer-out or transfer-in. Table 11 shows the household land self-cultivation characteristics for 218 land self-cultivation households or 40.4% of the households investigated. The average contracted land area per household was 0.25 ha; however, the medium- and low-level livelihood capital households were managing comparatively higher land areas than the high-level livelihood capital households. Notably, the LH-type households cultivated the most land (0.28 ha) and the HL-type households cultivated the least land (0.19 ha), indicating that poorer livelihood capital households could depend more on land. The differences of self-cultivated land areas were also smaller than the differences of transfer-out and transfer-in land areas, which was because the initial contracted land for the different households was distributed based on population; therefore, the household land area differences were not large when the land transfers were not considered.

However, due to the dispersed land in the study region, the single land plot areas were mostly less than 0.1 ha. The HH-type and HL-type households had no single plots greater than 0.2 ha, and few other household types had single land plots greater than 0.2 ha, indicating that the fragmentation of land was serious. The high-level and medium-level livelihood capital household paddy land areas were less than the low-level livelihood capital household paddy land areas. All household types, except for the HL-type households, had more non-irrigated land than paddy land, which suggested that cultivating non-irrigated land required less labor force and therefore released household members to work in non-farming livelihood activities. The different cultivated land area types in the high- and medium-level livelihood capital households were paddy land < irrigated land < non-irrigated land, but in the low-level livelihood capital households were non-irrigated land > paddy land > irrigated land, perhaps because the vegetable or cereal demands in the different households were different. For example, most smallholders often reclaimed a few plots of land in front of their houses or in their yards to grow vegetables, such as tomatoes, cucumbers, pumpkins, peppers, and cabbages, for daily food consumption. The LH-type and LL-type households, however, were more involved in cereal crop farming, mainly rice, on contracted paddy land.

The analysis of the reasons for households without transfer-out and -in land could indirectly clarify the drivers of household land self-cultivation. Most households generally cultivated a small area of land to ensure basic food supplies; however, for low-level livelihood households, this land was often their main source of income. The lack of appropriate land tenants restrained households with high-level and medium-level livelihood capital from transferring-out land; however, when disregarding the transfer-in land, the fact that the proportion of non-agricultural livelihood activities in the study area was close to 60% suggested that a high number of the rural labor force had flowed into non-agricultural sectors, which meant that because of the lack of a household labor force, households were unable to fully manage redundant land. In addition, low farm income, high expenditure, the lack of an appropriate landlord, and disputes were other reasons that households did not choose to transfer-in land, in turn suggesting that they could just cultivate the stable contracted land.

#### 3.4.4. Land Abandonment

Table 12 shows the main characteristics of household land abandonment. Around 70 households or 11.9% of the households investigated chose to abandon their land, with the average abandoned land area per household being 0.18 ha. However, only 17 households had abandoned all land and were not managing any land, suggesting that most of the rural households in the study areas correlated with land. Table 8 shows that the high-level and medium-level livelihood capital household land abandonment ratios were slightly higher than the low-level livelihood household ratios. Due to the livelihood diversity differences, the household land abandonment ratios were HL-type > HH-type, MH-type > ML-type, and LH-type = LL-type, and the average abandoned land area was low-level livelihood capital households > medium-level livelihood capital households > high-level livelihood capital households; therefore, the lower the household livelihood capital, the larger the abandoned land area. This was probably because households with poor livelihood capital did not have enough manpower to manage the land and did not have the social capital to transfer-out land.

The main common reason for land abandonment was found to be the inability to find an appropriate land tenant, possibly due to the lack of necessary land transfer market information. In addition, households, except for the ML-type, claimed that the low income from the land was also a driver, and except for the HH-type households, other households claimed that insufficient labor was a driver. Notably, households with low-level livelihood capital did not take the low annual rent of land transfer-out as a driver, which implied that they might have been more concerned about whether the land could be transferred-out, whereas households with high-level and medium-level livelihood capital were more likely to pay attention to whether the land could be transferred-out effectively; that is, “poorer” households appeared to care more about the transfer itself, but “rich” households cared more about profitable rent. Other drivers that prompted households to abandon surplus and inferior land were long production distances, fragmented land, imperfect agricultural facilities, poor land quality, lack of funds, and relevant production service constraints, and they only managed superior land that could support their daily rice and vegetable needs, indicating that reasons for land abandonment were diverse and differentiated.

## 4. Conclusions and Suggestions

### 4.1. Conclusions

Rural households appeared to adjust their land use decisions based on their livelihood endowments to optimize their comprehensive profits. Based on data from 540 households in Qionglai, Nanjiang, and Luxian in Sichuan Province, the rural household livelihood capital and diversity indices were calculated, after which the households were divided into six types using the combined indicators. A land use decision-making analysis framework was developed to explore the characteristics and relationships between household livelihood capital, livelihood diversity, livelihood strategies, and land use decision-making, and the main conclusions were as follows.

Overall, except for human capital, the household total livelihood capital was not high, with the natural capital, financial capital, physical capital, and social capital all being relatively low. The non-agricultural household livelihood strategy trends in the survey area were a microcosm of the effects of industrialization and urbanization in Sichuan Province and were similar to trends in other developing countries such as Vietnam [54], India [55], and Morocco [56]. Because of the potential non-farming opportunity differences between the regions, the households in Nanjiang and Qionglai were found to have more diverse livelihood activity choices than Luxian. As the disparities in livelihood capital and livelihood diversity expanded, the differences between the household livelihood strategies and living standards were growing, which could lead to imbalanced and inadequate development in many rural areas in Sichuan.

The investigated rural households were divided into HH-types, HL-types, MH-types, ML-types, LH-types, and LL-types using the dual indicators, the livelihood capital and livelihood diversity indices. The LL-type household was the largest household group (185, 34.3%). The distribution of the different household groups in Qionglai was well-proportioned but was inhomogeneous in Nanjiang and Luxian. Rural households with a high-level livelihood diversity, such as the HH-type, MH-type, and LH-type households, mostly chose to engage in diversified and low-risk livelihood activities; however, the rural households with low-level livelihood diversity, such as the HL-type, ML-type, and LL-type households, mostly participated in specialized and high-risk livelihood activities, which meant their livelihood strategies were differentiated and polarized.

Of the different household types in the different study regions, the households that adopted various land use decisions were characterized by self-cultivation households (218, 40.4%), land transfer-in households (159, 29.4%), land transfer-out households (143, 26.5%), and land abandonment households (70, 11.9%). Specifically, there were significantly more part-time smallholder households managing small areas of cultivated land to satisfy their daily rice and vegetable demands or households relying on farming for a portion of their income than professional farm households involved in large-scale agricultural land operations. However, a distinct modern agricultural production pattern was found, with a moderate land management scale gradually replacing fragmented and small-scale land management. To reduce expenditure and achieve comprehensive profits, mechanized and capital-intensive agricultural production was replacing the need for an intensive agricultural production labor force. However, whether smallholder agricultural management and operations were being adapted to modern and efficient agricultural production in Sichuan requires further exploration.

The land transfer rate in the study area was not high, with the land transfer-in and transfer-out ratios both being less than 30%. Households in the survey area were more inclined to transfer-in land than transfer-out land, and the transfer-in land scale (2.48 ha per household) was significantly greater than the transfer-out land scale (0.20 ha per household). During the land transfer process, it was found that there was a significant rise in non-food agricultural land use, which indicated that rural households were seeking to pursue higher financial benefits from agricultural production by growing high-value-added cash crops. Due to the incomplete land transfer market, landlords and tenants could not easily obtain related information about the land rents, procedures, or time periods, or establish long-term and guaranteed agreements involving land transfers, which meant that households tended to contract with familiar landlords and tenants, such as relatives or friends, which possibly restricted their ability to make economically optimal land transfer decisions.

Because of these common driving factors, such as the lack of appropriate land tenants, limited labor, and low farm income, some farm households chose to abandon their inferior land and only wanted to cultivate superior land or completely enter the non-agricultural sector; however, very few rural households (only 17 out of 540 households) had abandoned their land, indicating most of the households in the study area still correlated with the land. Even though many of the land abandonment households had left their agricultural land, they had not completely abandoned agricultural production, with many retaining a small plot of contracted land or idle farm tools to allow them to restart agricultural production in the future, perhaps because of the special earth-love-knot. Moreover, the COVID-19 pandemic significantly affected rural people who took urban employment; so, they who kept latent agricultural production could depend more on the social security function of the land.

Generally speaking, agricultural land was not found to be indispensable to survival for most rural households in the study area as they were more focused on the comprehensive development of their optimal livelihood activities, most of which were not dependent on their land for survival. Only households that were seeking agricultural production profits, such as experienced farmers and large planters, were found to rely on the land for survival and livelihood activity development. However, the low land transfer rate, the fragmented landholdings, the smallholder agricultural management, and the land abandonment in the study area revealed that the separation between the rural population and the agricultural land had not kept pace with urbanization, industrialization, or modernization, and had led to inefficient land use and discordant human–land relationships.

### 4.2. Suggestions

Rural household livelihood capital was the main reason for the choice of household livelihood strategy decision-making on land use. Therefore, based on the comprehensive analysis and conclusions regarding the different types of household livelihood capital, livelihood diversity, livelihood strategy, and land use decisions, the following suggestions are given.

First, human capital was found to be the core rural household capital in the study area and played an essential role in the household livelihood strategy choices and their decisions to choose diverse livelihood activities to reduce their livelihood risks and indirectly improve the other lower livelihood capitals. Therefore, because of these diverse rural livelihood strategies, policies should be targeted and differentiated. For example, for non-agriculture and non-agriculture-dependent households, related non-agricultural vocational education and employment training could be offered to improve their employment and income opportunities in non-agricultural industries. To advance agricultural product quality and increase the agricultural income and human and financial capital in rural households that have pure-agriculture and agriculture-dependent livelihood strategies, the government could improve and strengthen household cultivation, breeding, and sales skills by providing guidance on crop management, agricultural technology, and agricultural sales techniques.

Second, because of the adverse external factor effects in the study area on smallholder livelihood activities, such as the asymmetric agricultural production market information, the rudimentary agricultural facilities, and insufficient agricultural policies, focus is needed on improving living standards and optimizing land use decisions. Agricultural households, such as experienced farmers and large planters, are at risk of suffering from severe economic losses, falling agricultural production scales, and land abandonment to enter the non-agricultural sector. Therefore, measures are needed to guide agricultural households in choosing the appropriate planting structures based on market demand, and subsidies need to be increased to encourage local agriculture production and stimulate the enthusiasm of experienced farmers to be involved in moderate farming to make up for their low relevant capitals. The government also needs to improve agricultural infrastructures, such as ditches, roads, and drying fields; provide households with the necessary financial services; properly simplify the prerequisites for household loans and mortgages; and encourage farm households to invest in land for long-term agriculture operations.

Third, the efficient transfer of land and a moderate expansion of the land management scale could improve land use efficiency. While currently implemented land transfer policies partly influenced the household livelihood capital configurations, livelihood activity choices, and specific land use decision-making, these policies have not dealt with the correlations between the labor force transfer and land transfer. Therefore, to realize moderate agricultural scale production and orderly population mobility and urbanization, the government needs to implement further measures to promote efficient, fair, and reasonable land transfers and promote labor force transfers by developing sophisticated labor force and land transfer markets in rural regions based on a full consideration of the land area, plots, location, and labor force characteristics.

Fourth, generally, land abandonment occurs after households comprehensively assess their livelihood capital and livelihood abilities. As land abandonment can have both positive and negative effects on agricultural production, food security, and ecological recovery, inferior land (hard to cultivate and extremely low productivity) should be converted into forests and grasslands to protect the natural environment and realize ecological restoration. To safeguard food security and maintain the dynamic quantity and quality balance of cultivated land, support is also needed for land leveling and soil optimization to increase land productivity and prevent superior land from being abandoned or inefficiently used.

## Figures and Tables

**Figure 1 ijerph-19-09485-f001:**
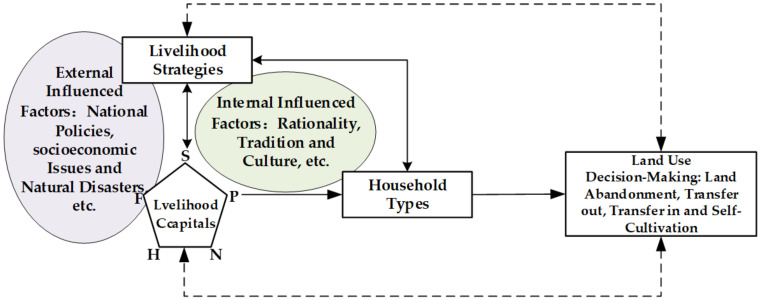
Framework for household land use decision-making based on the SLF.

**Figure 2 ijerph-19-09485-f002:**
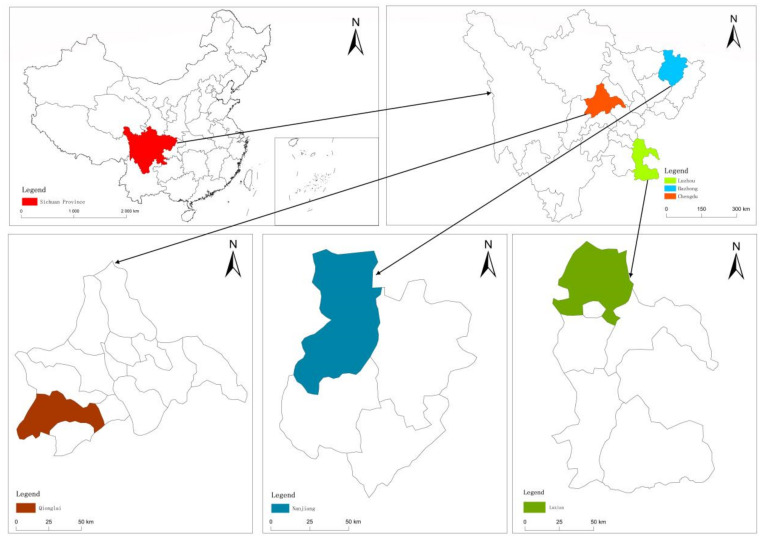
Study area.

**Figure 3 ijerph-19-09485-f003:**
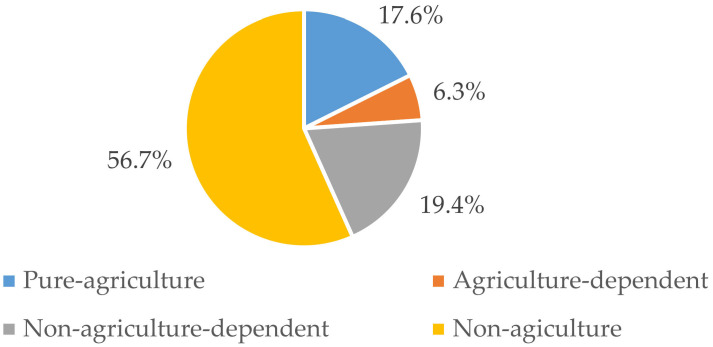
Ratio of household livelihood strategies in the study area.

**Figure 4 ijerph-19-09485-f004:**
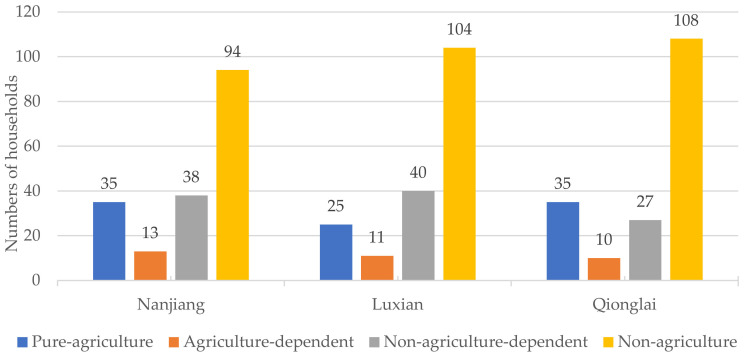
Distribution of household livelihood strategies in the study area.

**Figure 5 ijerph-19-09485-f005:**
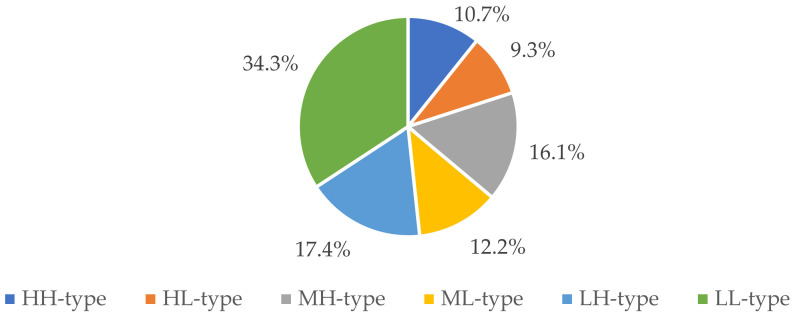
Ratios for the different household types in the study area.

**Figure 6 ijerph-19-09485-f006:**
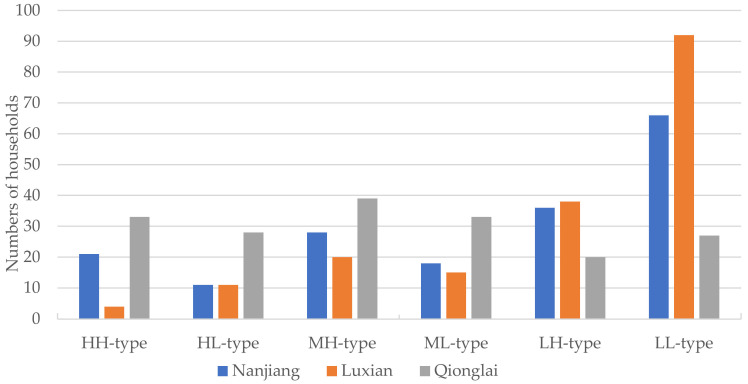
Distribution of the different household types in the study areas.

**Figure 7 ijerph-19-09485-f007:**
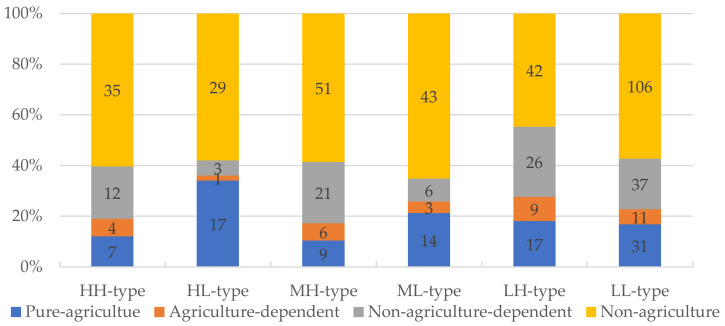
Ratio of the different household type livelihood strategies in the study area.

**Table 1 ijerph-19-09485-t001:** Evaluation index system for household livelihood capital.

Types	Index	Weight	Description and Assignment
Human capital	Proportion of labor force to the total population	0.0853	Male labor force (16–60 years old) Female labor force (16–55 years old)
	Per capita education attainment	0.0850	Education attainment per capita (year)
	Per capita degree of health	0.0813	Healthy = 1, healthy to some extent = 0.3, ill = 0
Natural capital	Per capita cultivated land area	0.0727	Actual operating cultivated land area per capita (ha/person)
	Per capita orchard land area	0.0530	Actual operating orchard land area per capita (ha/person)
	Per capita forest land area	0.0382	Actual operating forest land area per capita (ha/person)
Financial capital	Per capita annual income	0.0875	Income per capita in 2020 (CNY/person)
	Per capita cash and bank savings	0.0759	Sum of cash and bank deposits per capita in 2020 (CNY/person)
	Per capita loan funds	0.0480	Loan funds per capita in 2020 (CNY/person)
Physical capital	Per capita fixed assets	0.0773	Present value of the per capita durable goods, agricultural machinery, and transportation vehicles, etc., in 2020 (CNY/person)
	Per capita standardized rural housing area	0.0659	Rural housing area per household multiplied by the housing quality value (poor = 0.3, not bad = 0.7, superior = 1)
	Whether they have the urban commercial housing	0.0539	Yes = 1, no = 0
	Per capita livestock	0.0472	Chicken, duck = 0.2; pig = 0.5; and cattle, sheep = 1
Social capital	Members in public office	0.0637	Household members or relatives serving as township cadres or other public officials
	Per capita expenditure in maintaining social relations	0.0651	Expenditure sums on participating in weddings, funerals, and other social parties, etc., in 2020 (CNY)

**Table 2 ijerph-19-09485-t002:** Household income sources.

Primary Income Source	Subdivision Income Source
Working for others	Agriculture
	Secondary industries
	Tertiary industries
Agricultural operation	Agricultural planting
	Forestry
	Livestock and poultry breeding
Fixed salary	Public office, school, hospitals, etc.
Industrial and commercial operation	Industry, construction, mining industry
	Transportation, post and telecommunications industry
	Wholesale and retail trade, catering industry
	Social service, culture, education, and health industry
Other industry	Other industry

**Table 3 ijerph-19-09485-t003:** Household livelihood capital values in the study areas.

Type	Nanjiang	Luxian	Qionglai	Mean
Human capital	0.147	0.132	0.177	0.152
Natural capital	0.001	0.001	0.005	0.002
Financial capital	0.006	0.006	0.008	0.007
Physical capital	0.013	0.012	0.017	0.014
Social capital	0.011	0.006	0.016	0.011
Total	0.178	0.157	0.223	0.186

**Table 4 ijerph-19-09485-t004:** Household livelihood diversity index in the different study areas.

Study Area	Nanjiang	Luxian	Qionglai	Mean
Livelihood Diversity Index	0.230	0.168	0.248	0.215

**Table 5 ijerph-19-09485-t005:** Primary household livelihood characteristics.

Livelihood Characteristics	Household Type					
	HH-Type	HL-Type	MH-Type	ML-Type	LH-Type	LL-Type
Labor force per household	3.34	2.66	3.08	2.45	2.15	1.16
Per capita education attainment (Year)	9.47	8.23	7.91	7.66	5.75	5.20
Annual income per household (CNY)	118,300	174,400	98,300	97,700	67,800	47,200
Per capita standard housing area (m^2^)	62.40	63.52	50.65	67.91	48.59	60.93
Per capita cultivated land area (ha)	0.099	1.477	0.079	0.310	0.084	0.109
Per capita forest area (ha)	0.065	0.155	0.009	0.052	0.027	0.032
Per capita orchard area (ha)	0.047	0.080	0.162	0.028	0.008	0.012

**Table 6 ijerph-19-09485-t006:** Household livelihood capital and livelihood diversity index.

Household Type	Ratio	Livelihood Capital						Livelihood Diversity Index
		H	N	F	P	S	Total	
HH-type	10.7%	0.2017	0.0020	0.0104	0.0297	0.0297	0.2735	0.483
HL-type	9.3%	0.1916	0.0083	0.0192	0.0374	0.0250	0.2815	0.041
MH-type	16.1%	0.1792	0.0025	0.0058	0.0095	0.0138	0.2108	0.454
ML-type	12.2%	0.1776	0.0021	0.0076	0.0134	0.0085	0.2092	0.034
LH-type	17.4%	0.1332	0.0008	0.0035	0.0083	0.0051	0.1509	0.421
LL-type	34.3%	0.1134	0.0010	0.0041	0.0082	0.0052	0.1319	0.027
Mean		0.1661	0.0028	0.0084	0.0178	0.0146	0.2096	0.243

**Table 7 ijerph-19-09485-t007:** Land use decisions by households in the different study areas.

Land Use Decision		Study Area		
		Nanjiang	Luxian	Qionglai
Transfer-out	Household	27	51	65
	Area (ha)	0.21	0.13	0.25
	Income (CNY/ha per year)	6925	4955	10,224
Transfer-in	Household	78	56	25
	Area (ha)	0.48	0.24	13.27
	Expenditure (CNY/ha per year)	3328	2985	8537
Self-cultivation	Household	67	86	65
	Area (ha)	0.28	0.26	0.20
Abandonment	Household	38	17	15
	Area (ha)	0.25	0.06	0.10

**Table 8 ijerph-19-09485-t008:** Ratio of different land use household decisions.

Household Type	Land Use Decision				
	Transfer-Out (%)	Transfer-In (%)	Self-Cultivation (%)	Abandonment (%)	Total (%)
HH-type	27.6	20.7	36.2	15.5	100
HL-type	27.8	31.5	24.1	16.7	100
MH-type	22.6	22.6	38.7	16.1	100
ML-type	24.3	21.6	40.5	13.5	100
LH-type	22.7	26.4	38.2	12.7	100
LL-type	21.1	23.9	42.3	12.7	100

**Table 9 ijerph-19-09485-t009:** Land transfer-out characteristics.

Household Type		HH	HL	MH	ML	LH	LL
Household		17	15	21	18	25	45
Area (ha)		0.214	0.277	0.230	0.224	0.122	0.180
Income (CNY/ha per year)		10,776	10,328	9463	9761	7418	8567
Tenants of the land transfer-out	Relatives (%)	11.11	14.29	20	5.56	7.69	10.64
	Large planters (%)	44.44	35.71	35	38.89	19.23	21.28
	Other villagers in the same village (%)	16.67	42.86	25	22.22	19.23	36.17
	Villagers in other village (%)	5.56	0	10	16.67	23.08	2.13
	Cooperative (%)	11.11	0	10	0	3.85	6.38
	Agriculture enterprise (%)	0	7.14	0	5.56	15.38	10.64
	Others (%)	11.11	0	0	11.11	11.54	12.77
Total (%)		100	100	100	100	100	100
Reasons for land transfer-out	Poor land quality (%)	0	0	9.38	4.35	0	1.96
	Small land scale (%)	10.71	0	12.5	4.35	6.67	1.96
	Insufficient labor force (%)	35.71	60	28.13	43.48	26.67	33.33
	Low land income (%)	25	30	25	21.74	16.67	13.73
	Uniform land transfer-out inside the village (%)	28.57	10	21.88	17.39	33.33	31.37
	Others (%)	0	0	3.13	8.7	16.67	17.65
Total (%)		100	100	100	100	100	100

**Table 10 ijerph-19-09485-t010:** Land transfer-in characteristics.

Household Type		HH	HL	MH	ML	LH	LL
Household		12	17	23	19	32	56
Area (ha)		1.48	13.65	0.71	4.63	0.34	0.30
Expenditure (CNY/ha per year)		3642	7463	6418	10,269	6149	5134
Landlord of transfer-in	Relatives (%)	25	23.81	50	27.27	31.25	37.29
	Villagers in the same village (%)	58.33	57.14	50	63.64	65.63	57.63
	Villagers in other villages (%)	8.33	19.05	0	9.09	3.13	1.69
	Others (%)	8.33	0	0	0	0	3.39
Total (%)		100	100	100	100	100	100
Purpose of transfer-in	Food crop (%)	35.71	45	75	65	72.22	70.49
	Non-food crop (%)	57.14	45	20.83	30	22.22	29.51
	Agriculture enterprise (%)	0	5	4.17	5	0	0
	Others (%)	7.14	5	0	0	5.56	0
Total (%)		100	100	100	100	100	100
Cause of transfer-in	Sufficient labor (%)	10	12.5	10	4.76	8.11	3.45
	Friends and relatives’ requirements (%)	30	18.75	40	38.1	37.84	53.45
	Sizable economic benefits (%)	40	50	20	47.62	32.43	25.86
	Others (%)	20	18.75	30	9.52	21.62	17.24
Total (%)		100	100	100	100	100	100

**Table 11 ijerph-19-09485-t011:** Land self-cultivation characteristics.

Household Type			HH	HL	MH	ML	LH	LL
Household			20	13	33	27	39	86
Area (ha)			0.20	0.19	0.27	0.23	0.28	0.24
Cultivated land use structure	Plot	<0.1 ha	7.4	8	4.9	6.37	7.28	7.8
		0.1≤ and <0.2 ha	0.5	0.23	0.85	0.59	0.85	0.64
		0.2≤ and <0.3 ha	0	0	0.18	0.1	0.08	0.08
		≥0.3 ha	0	0	0.15	0.03	0.03	0.07
	Type	Paddy land (ha)	0.04	0.05	0.05	0.06	0.09	0.10
		Irrigated land (ha)	0.05	0.08	0.08	0.03	0.04	0.02
		Non-irrigated land (ha)	0.11	0.07	0.15	0.14	0.15	0.13
Reasons for not transferring-out land		Main income source from land (%)	15	14.29	10	30	28.3	26.61
		Meet the basic food demand (%)	55	33.33	42.5	40	43.4	52.42
		Low annual rent (%)	5	19.05	7.5	0	7.55	4.84
		Lack of the appropriate tenant (%)	25	28.57	37.5	30	16.98	12.9
		Worry about disputes (%)	0	0	0	0	1.89	0.81
		Others (%)	0	4.76	2.5	0	1.89	2.42
Total (%)			100	100	100	100	100	100
Reasons for not transferring-in land		Low income from land (%)	28.57	26.09	34.15	31.58	28.3	24.17
		Limited labor (%)	61.9	43.48	46.34	50	45.28	56.67
		High expenditure (%)	0	13.04	4.88	7.89	11.32	9.17
		Lack of the appropriate landlord (%)	4.76	17.39	4.88	5.26	9.43	6.67
		Worry about disputes (%)	0	0	4.88	5.26	3.77	3.33
		Others (%)	4.76	0	4.88	0	1.89	0
Total (%)			100	100	100	100	100	100

**Table 12 ijerph-19-09485-t012:** Land abandonment characteristics.

Household Type		HH	HL	MH	ML	LH	LL
Household		7	9	10	7	13	24
Households without cultivating any land		4	3	2	2	3	3
Area (ha)		0.18	0.10	0.26	0.06	0.20	0.20
Reasons for land abandonment	Low income from land (%)	17.86	9.88	12.68	0	5.35	9.09
	Insufficient labor force (%)	0	9.88	31.69	21.43	26.75	18.18
	Low annual rent of land transfer-out (%)	17.86	19.75	6.32	21.43	0	0
	Lack of appropriate land tenants (%)	28.57	17.28	21.13	14.28	30.86	27.27
	Others (%)	35.71	43.21	28.17	42.86	37.04	45.46
Total (%)		100	100	100	100	100	100

## Data Availability

Data available on request.

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
