# Peer review of "Household Groups’ Land Use Decisions Investigation Based on Perspective of Livelihood Heterogeneity in Sichuan Province, China"

_ijerph, 2022, doi:10.3390/ijerph19159485_

Round 1
Reviewer 1 Report
2022.06.27 Review
Household Groups’ Land Use Decisions Investigation Based on Perspective of Livelihood heterogeneity in Sichuan Province, China. Tang et al.
Overall assessment.
This research makes an important contribution to agricultural and rural livelihoods literature. The authors are to be commended for there work in this area. The Sustainable Livelihoods Framework is especially relevant to the research questions asked and the search for better understanding of the diversity of incomes rural households use to support their families. Good literature review and set up. The extensive interviews and effort to quantify and find significance in the data was a good way to begin to categorize and explore livelihood heterogeneity. It is important to note it is still a purposeful sample and can’t be generalized to other rural places beyond Sichuan. However the random sampling of villages and households in the Sichuan Province offer excellent representation of rural households in this province. It was a good application of the SLF. The findings section is written quite densely (use of the English language a major stumbling block) and it is hard to follow. The analysis of findings in Section 4 presented some useful observations that future research can leverage in Conclusions and Suggestions. The land transfer and abandon issues are not trivial, small landholders who derive even only a small portion of their livelihoods from agriculture may benefit greatly if the household is more likely to be food secure even though financially poor. Recognition of the value of the human-land relationship was insightful (special earth-love-knot); working the land is not simply an occupation to provide household income, there are social-emotional-and environmental human relationships involved that influence occupational and income diversity choices. This paper raises more questions than it answers, I hope the authors continue in this line of research.
Several observations.
1. The next to last paragraph in Section 1 (line 124-128) claims that Sichuan Province that was purposefully selected as “representative” and “typical” of rural China.
There is a need for national and subregional/provincial data to substantiate this claim. What is the average rural farm size? (hectare), cropping systems (some crops have higher cash value than others), household size, distances to urban centers (and off-farm employment) , rural population density, etc in China compared to Sichuan? This would strengthen the claim of “typical” and representative.
2. Three levels of data-individual, community and region are gathered via survey and semi-structured interviews. The authors have a lot of great data, only a portion of which were used in this paper . Consider doing a multilevel analysis as the next step in future work.
3. In the materials and methods section, need more detail on the how the surveys were developed and pilot tested. What constitutes a specialized household with big scale land vs smallholders? How was size and crops defined as specialized? Monocropping? Rice and aquaculture Vs garden farming? More detail regarding the agricultural systems would inform land use decisions.
4. How were data processed? Other researchers would be interested in knowing how the semi-structured interview transcripts were coded, (NVIVO? Other software? Or by hand? 540 households are a lot to code by hand) by how many people, and how data were reconciled to reduce coding bias. How were the surveys analyzed? What software programs (surveys and semi-structured data)?
5. Findings. Data are primarily presented descriptively. The reader would benefit from an explanation of the coefficients reported in the tables and their meanings, significance of mean differences/similarities
6. English language is a big problem in this paper
Although I have noted there are a number shortfalls with this paper,
I would like to see this paper published. There is a paucity of literature in this area and authors findings have much to contribute. The English word choices, sentence grammar and paper structure are MAJOR limitations to this work. Poorly constructed sentences and use of English were major obstacles in reading and understanding what they authors were trying to convey. I do understand English is not the author’s native language, however a English editor is a MUST for publishing this paper and will require substantive work.
I like this work very much and hope that the authors will invest in a good editor; and wish them success as they continue to analyze their data and develop their theory.
Reviewer 2 Report
This paper chose a very meaningful question. The discussion of land use decision is of great significance to the study of expanding the livelihood strategy of Chinese households, and the data is adequate and reliable. However, the overall writing lacks substantial theoretical contributions and is more like a solid research report than a thesis.
1.The dialogue between the article and the Sustainable Livelihoods Framework (SLF) is not enough. Based on Figure 1 and the corresponding description, the reader is not able to get the answers they are looking for, namely why this framework can be applied to China, or what is special about the China scenario of this framework and whether it needs to be supplemented? If there is a more appropriate expression here, for example, FIG. 1 can be combined with China's national conditions to improve the mature framework in the world and bring second-level or even third-level measurement indicators, then the rationality of the framework will be improved.
2.In practice, the assumptions of rational people are not entirely consistent with the livelihood behavior of families. Therefore, "whether the hypothesis of rational man can explain the livelihood behavior of households" is itself a proposition worth exploring. However, this paper does not understand this proposition, which is an important theoretical differentiation point in farmer research. Schultz, Chayanov and others have been debating the issue for a long time.
3.Based on Comment 2, even though the paper gives a detailed description and classification of the surveyed households, the contents of the discussion are likely to be inconsistent with the actual logic of the households. For example, it is common for smallholder farmers to be involved in land transfers more than families. However, influenced by neoliberalism, this paper does not fully recognize the rich value of land to small producers from the perspective of practice and livelihood. In fact, small-scale producers have been playing a key role in transforming the global food system. Their contribution to food supply, rural social stability and cultural heritage, conservation of species diversity and adaptation to climate change is outstanding.
Round 2
Reviewer 2 Report
The response of the manuscript is basically complete, and the LUDF framework can be better used to solve the research problem.